# Outcome of Monochorionic Diamniotic Twins with Twin Reversed Arterial Perfusion Sequence: Interstitial Laser versus Endoscopic Cord Occlusion

**DOI:** 10.3390/jcm12206593

**Published:** 2023-10-18

**Authors:** Izabela Walasik, Magdalena Litwinska, Katarzyna Janiak, Krzysztof Szaflik, Piotr Kaczmarek, Artur Ludwin, Ewelina Litwinska

**Affiliations:** 1I Department of Obstetrics and Gynecology, Medical University of Warsaw, 02-091 Warsaw, Poland; izabela.a.walasik@gmail.com (I.W.); litwinska.magdalena@gmail.com (M.L.); ludwin@cm-uj.krakow.pl (A.L.); 2Department of Gynecology, Fertility and Fetal Surgery, Polish Mother’s Memorial Hospital in Lodz, 93-338 Lodz, Poland; kasiajaniak@me.com (K.J.); krzysztofszaflik@wp.pl (K.S.); kaczmarekpiotr1@gmail.com (P.K.)

**Keywords:** twin reversed arterial perfusion sequence, TRAP pregnancy, monochorionic twins, prenatal interventions

## Abstract

Twin reversed arterial perfusion sequence (TRAP) is a rare complication of monochorionic twins (MC). This study aimed to describe and compare the short- and long-term outcomes of MC pregnancies with the TRAP sequence treated with two different techniques: interstitial fetal laser (IFL) (*n* = 22) versus endoscopic cord occlusion (CO) (*n* = 24). The study population included 46 MC pregnancies with TRAP. Pregnancy loss within 2 weeks after the procedure occurred in 27% of cases (6/22) in the group treated with IFL and in 8% of cases (2/24) in the group treated with CO. The survival rate of the pump twin was 73% (16/22) in the IFL group and 83% (20/24) in the group treated with CO. The median gestational age at birth was 38 weeks in the group treated with IFL and 35 weeks in the group treated with CO. The rate of preterm birth before 34 weeks was 12.5% (2/16) in the group treated with IFL and 32% (7/22) in the group treated with CO. In the group treated with IFL, there were no cases of neurological disabilities reported by the parents compared to three cases in the CO group. IFL is associated with a higher risk of early pregnancy loss; however, if the pregnancy progresses, it is associated with lower risks of preterm birth and neurological disabilities in the survivors.

## 1. Introduction

Twin reversed arterial perfusion sequence (TRAP) is a rare complication of monochorionic pregnancy (MC), which occurs in 1 per every 35,000 deliveries (1–3% of MC pregnancies) [1,2,3]. The arterio-arterial anastomoses in the placenta lead to the reverse transportation of blood from the pump twin to the acardiac fetus [4,5]. The literature reports a 50% mortality rate for the pump twin due to high-output cardiac failure or preterm labor, if left untreated [6]. 

The prenatal management of TRAP has evolved from highly invasive interventions involving hysterotomy and the removal of the acardiac twin [7,8,9] to less invasive procedures aiming to interrupt the cord blood flow to the acardiac fetus [10,11,12]. These include endoscopic procedures such as the insertion of cord coils, ligation of the umbilical cord, laser coagulation of placental anastomoses and monopolar or bipolar diathermy of vessels within the acardiac twin cord, ultrasound-guided ablation of intrafetal vessels via the injection of alcohol, interstitial laser or radiofrequency, as well as newer techniques like high-intensity focused ultrasound. The reported data vary in terms of the method and timing of the procedure; however, there is an increasing amount of evidence in favor of prophylactic intervention before 16 weeks’ gestation [13,14,15,16]. Also, the vast majority of the reports are limited to short-term effects. However, the assessment of long-term outcomes including neurological development in the survivors is crucial in accurate patient counselling. It has been advocated that the pump twin in the TRAP sequence is at a high risk of both cardiac and neurodevelopmental complications due to chronic hemodynamic imbalances [17,18].

Our study aimed to describe and compare the short- and long-term outcomes of MCDA twin pregnancies complicated by the TRAP sequence treated with two different minimally invasive techniques: interstitial fetal laser (IFL) versus endoscopic cord occlusion (CO).

## 2. Materials and Methods

This was a retrospective study of 46 MCDA pregnancies diagnosed with TRAP sequence in two fetal surgery centers in Poland between 2013 and 2020. The diagnosis of TRAP was confirmed in all cases by demonstrating an anatomically normal fetus coexisting with a morphologically abnormal twin without a heart structure in MCDA pregnancy. The retrograde perfusion from the pump twin to the acardiac twin was shown using color flow Doppler. Ultrasound scans were performed by certified fetal medicine specialists in order to assess the biometry and detect possible abnormalities. Gestational age was calculated on the basis of the pump twins’ crown–rump length (CRL), followed by weekly assessment of growth and hemodynamic function. Fetal cardiac compromise was defined qualitatively by fetal echocardiographic changes including ventricular hypertrophy, depressed ventricular contractility and valve regurgitation. 

Study population was divided into two groups: MCDA twins treated with IFL at 12–22 weeks’ gestation (*n* = 22) and MCDA twins treated with CO at 17–27 weeks’ gestation (*n* = 24). In both centers, fetal surgery was offered in all cases with confirmed diagnosis of TRAP sequence. 

Interstitial laser coagulation of the pelvic vessels was performed under ultrasound guidance. Ultrasound scanning was used to obtain the transverse section of the lower fetal abdomen of the acardiac twin and to define the appropriate site of entry on the maternal abdomen. The procedure was carried out under maternal local anesthesia (10 mL of 1% lignocaine) with intravenous sedation. Under continuous ultrasound guidance, an 18-gauge needle was introduced transabdominally into the amniotic cavity. Color flow mapping was then used to identify fetal pelvic vessels, and the needle was positioned close to it. A laser fiber (0.7 mm in diameter) was then inserted through the needle with its tip pointing towards the pelvic vessels. The coagulation was performed using an Nd:YAG laser (Dornier Med Tech, Wessling, Germany) continuously for 5–10 s using an output of 30–40 watts (W). The procedure was repeated until the absence of blood flow using color Doppler was confirmed. 

Laser coagulation of the umbilical cord was performed under endoscopic guidance. Ultrasound scanning was used to localize the placental insertion of the umbilical cords and to choose an appropriate site of entry on the maternal abdomen. The procedure was carried out under maternal local anesthesia (10 mL of 1% lignocaine) with intravenous sedation. Under continuous ultrasound guidance, a rigid 3 mm diameter fetoscope (Karl Storz, Tuttlingen, Germany) was introduced transabdominally into the cavity of the acardiac twin. If necessary, in order to improve the visualization, Ringer lactate solution was infused. A 0.7 mm laser fiber was then passed down to 1 cm beyond the tip of the fetoscope. The umbilical cord was than targeted, and the Nd:YAG laser (Dornier Med Tech, Wessling, Germany) was administered using an output of 30–40 W at a distance of 1 cm. The procedure was repeated until the absence of blood flow using color Doppler was confirmed.

The technique of intervention was chosen for several reasons. Intrafetal laser was a preferred technique in diagnosed cases and referred to the centers before 22 weeks’ gestation. Laser coagulation of the umbilical cord was performed in cases referred to the centers after 22 weeks’ gestation. The choice of the technique was also dependent on the technical issues, which included the position of the placenta, position of the TRAP fetus and patient’s body mass index (BMI). Cord occlusion was a preferred technique in patients with high BMI, resulting in poor visualization under ultrasound guidance. In cases of anterior placenta resulting in difficult access to the umbilical cord of the TRAP fetus, IFL was a preferred method. 

Perioperative tocolysis was provided via betamimetics if the gestation was >24 weeks. All patients received antibiotic prophylaxis (Penicillin 1.2 g IV). 

Maternal characteristics, procedure details and outcomes of pregnancy were collected from the databases of the centers. Preterm delivery (PTD) was defined as delivery occurring before 34 weeks of gestation. Preterm premature rupture of membranes (PROM) was classified as membrane rupture before 37 weeks of gestation. Children’s neurological statuses were assessed via questionnaires or telephone contact up to 36 months after the delivery. Developmental impairment was defined as conditions affecting children’s behavioral functioning, learning and physical development. Impairments include sensory status, epilepsy, autism and cerebral palsy. The developmental impairments and other abnormalities were reported by parents after pediatric consultations during regular checkups.

### Statistical Analysis

The χ^2^ and Fisher exact tests were used to compare categorical variables with Mann–Whitney test for continuous variables. All values are represented as mean ± SD, and *p*-value of <0.05 was considered statistically significant.

## 3. Results

A total of 46 patients with MCDA pregnancies complicated by the TRAP sequence were included in the study (Table 1). The study population was divided into two groups: pregnancies treated with interstitial laser (*n* = 22) and pregnancies treated with endoscopic cord occlusion (*n* = 24). Fetal cardiac compromise was described in 18% (4/22) of cases treated with laser and in 38% (9/24) of cases treated with cord occlusion. Fetal surgery was offered in all cases with a confirmed diagnosis of TRAP sequence.

Median gestational age at intervention was 16 weeks (range 12–22) in the group treated with lFL and 21 weeks (range 17–27) in the group treated with CO. In the group treated with IFL, in three cases, there was a need to repeat the procedure due to residual flow within the acardiac twin that was visualized one week after the surgery. In the group treated with CO, in two cases, the procedure failed due to poor visibility, and bipolar cord occlusion was conducted under ultrasound guidance. The pregnancy outcomes are included in Table 2.

Pregnancy loss within 2 weeks after the procedure occurred in 27% of cases (6/22) in the group treated with IFL and in 8% of cases (2/24) in the group treated with CO. The survival rate of the pump twin was 73% (16/22) in the group treated with IFL and 83% (20/24) in the group treated with CO. The median gestational age at birth was 38 weeks (range 32–41) in the group treated with IFL and 35 weeks (range 25–41) in the group treated with CO. The rate of preterm birth before 34 weeks was 12.5% (2/16) in the group treated with IFL and 32% (7/22) in the group treated with CO. The rate of birth before 37 weeks of gestation was three times higher in the cord occlusion group (63.6% vs. 18.8% *p* = 0.009). In the group treated with CO, there were two cases of extremely preterm birth at 25–27 weeks, and the neonates died a few hours after delivery due to severe prematurity. 

In the group treated with IFL, there were no cases of neurological disabilities reported by the parents at the minimum age of 24 months, with all children meeting their milestones. In the group treated with CO, there were three cases of neurological disabilities described at the age of 36 months that included severely delayed speech development, bilateral hearing impairment and cerebral palsy. In the case of a child with severely delayed speech development, an uncomplicated intrauterine intervention was performed at 22 weeks. The child was born at 38 weeks with a birthweight of 3000 g and received 9 pointes in the Apgar score. In the case of a child with bilateral hearing impairment, an uncomplicated procedure was performed at 21 weeks. The mother went into spontaneous labor, and the child was born at 32 weeks with a birthweight of 1870 g and received 8 points in the Apgar score. The neonate had respiratory distress and required a surfactant. The Auditory Brainstem Response test showed a high degree of bilateral hearing impairment. In the case of a child with cerebral palsy, the preterm premature rupture of membranes occurred at 32 weeks, 9 weeks after the intrauterine intervention, and the neonate was delivered at 33 weeks with a birthweight of 1700 g and received 8 points in the Apgar score. The child is now 3 years old and has been diagnosed with cerebral palsy, which severely affects their psychomotor activities. In all three cases, at the time of the intervention, the pump twin had signs of cardiac overload. 

## 4. Discussion

In this group of 46 MCDA pregnancies complicated by the TRAP sequence, we carried out two types of intrauterine interventions: ultrasound-guided interstitial laser ablation of pelvic vessels within the acardiac fetus and endoscopic laser ablation of the umbilical cord of the acardiac fetus. 

In our series, we offered intrauterine intervention in all cases diagnosed with TRAP sequence regardless of the presence or absence of cardiac overload in the pump twin. A study by Chaveeva et al. demonstrated that a delay in the intervention between the diagnosis of the TRAP sequence at 11–13 weeks until 16–18 weeks is associated with the spontaneous cessation of flow in the acardiac twin in 60% of cases; however, in about 60% of these, there is also death or brain damage in the pump twin [14]. Pagani et al. presented a series of six pregnancies complicated by TRAP without an immediate mass growth or hyperdynamic circulation and showed that a delay in the intervention is associated with a high risk of intrauterine death [19].

Both methods of treatment have their technical limitations. In three cases, IFL failed at the first attempt, and the procedure was repeated and led to the cessation of flow within the acardiac fetus. In all of these cases, the procedure was conducted after 16 weeks’ gestation, and the abdominal circumference of the acardiac twin was equal to or greater than that of the pump twin (acardiac/pump twin ratio ≥ 1, taken at the level of the stomach). In two cases, the CO procedure failed due to poor visibility, and bipolar cord occlusion was conducted under ultrasound guidance.

Both methods of treatment varied in terms of perinatal outcomes. In our study, the rate of preterm birth (<37 weeks of gestation) was almost three times higher in the CO group compared to the IFL group (63.6% vs. 18.8%; *p* = 0.09). Preterm birth is the strongest risk factor of impaired neurological function in children. In our study, we reported three cases of severe neurological disabilities that included severely delayed speech development, bilateral hearing impairment and cerebral palsy; all of them were reported in the group treated with CO. In the group treated with IFL, there were no cases of neurological disabilities in the children. 

The strengths of the study are that it provides a substantial addition to the limited number of prenatally treated cases reported in the literature and the provision of data on the neurological follow-up of the patients. The major limitations of the study are related to its retrospective design, the small size of the group and the lack of controls. These limitations result from the rarity of the condition, with a reported incidence of 1 in 35,000 pregnancies.

Multiple therapeutic options for TRAP sequence have been presented in the literature, but no strict guidelines for treatment have been established. In our study, we compared two techniques: IFL and CO.

Intrafetal laser is performed under ultrasound guidance and is based on ablating pelvic vessels in the acardiac twin. This appears to be the least invasive option with the use of an 18-gauge needle. Chaveeva et al., in a series of 104 TRAP pregnancies treated with IFL, reported a 76% survival rate for the pump twin [14]. Similar results were presented by Pagani et al., who reported a survival rate of 80% (42/51). A recent systematic review including 156 pregnancies complicated by the TRAP sequence and treated with IFL reported a 79% survival rate [20]. Our results are consistent with the previously published data, with a 73% survival rate of the pump twin (16/22) in the group treated with IFL. 

Cord occlusion techniques are performed under fetoscopy or ultrasound guidance [21,22]. This technique involves instruments with a larger diameter (3–4 mm), which increases the risk of membrane damage and PROM. This technique is also affected by numerous factors, which include the localization of the umbilical cord that determines the entry side as well as the translucency of the amniotic fluid. Lewi et al. reported a 75% (14/19) survival rate of the pump twin after treatment with cord coagulation with laser or bipolar cord occlusion in complicated MC pregnancies. [23]. Our study presents similar findings, with an 83% survival rate of the pump twin (20/24) in the cord occlusion group.

The comparison of both methods of treatment should not be limited to the survival rate, but should also take into account the gestational age at delivery as well as the long-term neurological outcomes of the survivors. In our study, the gestational age at delivery was significantly higher in the group treated with IFL compared to the CO group. Similarly, in a large study presented by Tan et al., intrafetal ablation was associated with a later median gestational age at delivery compared to the cord occlusion group (37 vs. 32 weeks; *p* = 0.04) [24].

The data on the long-term follow-up after intrauterine interventions for TRAP are limited. Several reports proved radiofrequency ablation (RFA) in TRAP as a safe and effective method of treatment. The pooled estimated survival rate of the pump twin treated with RFA was 80.8% (72.3–87.1) [14]. Similarly, Sugibayashi et al. reported an overall survival rate of the pump twin of 85% after RFA [25]. Also, there were no cases of neurological disabilities in the survivors [20]. However, Pagani et al. showed an increased risk of pPROM when using RFA compared to intrafetal options (22% vs. 7% *p* = 0.045) [19]. The data on the neurological status of the survivors after IFL or cord coagulation in the TRAP sequence is limited to the study by Lewi et al., who documented pregnancy and infant outcomes after laser or bipolar coagulation in monochorionic multiple pregnancies including TRAP cases. The authors concluded that neurological impairment was more frequent if the intervention was conducted after 23 weeks’ gestation and suggested that earlier intervention could reduce blood imbalance and provide proper neurological development [23]. Our study presented similar results, with all three cases of neurological impairment postnatally presenting cardiac compromise at the time of prenatal intervention.

### Implications for Clinical Practice

The prenatal diagnosis of TRAP is usually made at a routine first-trimester ultrasound scan. It is now well accepted that the intervention should not be delayed until 16 weeks due to high risk of demise for the pump twin. In our series, the survival rate was higher in the group treated with CO compared to the IFL group; however, survival does not always mean success, which is defined as a term birth of a neurologically healthy child. The major risk factor of neurological impairment is still preterm delivery. The rate of preterm birth before 34 weeks was nearly three times higher in the group treated with CO compared to the group treated with IFL (32% vs. 12.5%; *p* = 0.254).

The pathophysiology of the TRAP sequence is based on chronic hemodynamic imbalances, which form another risk factor of neurological disabilities in the survivors. In this view, an earlier intervention prevents not only the death of the pump twin, but also neurological disabilities. 

One of the most significant risks associated with intrauterine procedures is membrane rupture. Intrafetal laser, which requires an 18-gauge needle, carries a lower risk of membrane damage compared to fetoscopic cord occlusion. However, the use of IFL is limited by the size of the targeted vessels and, therefore, the gestational age at intervention.

It could be therefore advocated that the first trimester diagnosis of TRAP sequence should lead to an immediate intrauterine intervention using intrafetal laser. If, on the other hand, the diagnosis is made at a later gestational age, endoscopic cord occlusion should be considered as an option of management.

## 5. Conclusions

Interstitial laser is associated with a higher risk of early pregnancy loss; however, if the pregnancy progresses, it is associated with lower risks of preterm birth and neurological disabilities among survivors.

## Figures and Tables

**Table 1 jcm-12-06593-t001:** Characteristics of the study group.

	Interstitial Laser % (*n* = 22)	Cord Occlusion % (*n* = 24)	*p*
Gestational age at intervention (median) [SD] (weeks)	16 (2.44) (12–22)	21 (2.26) (17–27)	<0.05
Gestational age at presentation (median) [SD] (weeks)	15 (2.26) (11–21)	20 (2.44) (16–27)	<0.05
Cardiac failure for the pump twin	18 (4/22)	38 (9/24)	0.197
AC ratio of acardiac fetus/pump twin >1	13.6 (3/22)	33 (8/24)	0.171
Polyhydramnios at presentation	9 (2/22)	12.5 (3/24)	1.00
Operation time (mean) [SD] (minutes)	10 (1.83)	25 (5.82)	<0.05
Total energy used (mean) [SD] (Watt)	25 (10.34)	35 (6.39)	<0.05
Need for repeated procedure	13.6 (3/22)	8.3 (2/24)	0.659

**Table 2 jcm-12-06593-t002:** Pregnancy outcomes.

	Interstitial Laser % (*n* = 22)	Cord Occlusion % (*n* = 24)	*p*
Intrauterine death (*n*)	27.3 (6/22)	8 (2/24)	0.128
Gestational age at birth (median) [SD] (weeks)	38 [2.45] (32–41)	35 [3.1] (25–40)	0.005
Preterm delivery < 34 w	12.5 (2/16)	31.8 (7/22)	0.245
Preterm delivery < 37 w	18.6 (3/16)	63.6 (14/22)	0.009
Birthweight (mean) [SD] (grams)	3150 (749) (1780–3890)	2200 (409) (1000–3900)	<0.05
Neurological deficits in the survivors	0 (0/16)	14.3 (3/21)	0.243

## Data Availability

The datasets used and analyzed during the current study are available from the corresponding authors upon reasonable request.

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
