# Peer review of "Outcome of Monochorionic Diamniotic Twins with Twin Reversed Arterial Perfusion Sequence: Interstitial Laser versus Endoscopic Cord Occlusion"

_jcm, 2023, doi:10.3390/jcm12206593_

Round 1
Reviewer 1 Report
This paper described the outcomes of TRAP sequence treated by IFL or CO. IFL is associated with a high risk of early pregnancy loss, while CO is associated with a high risk of preterm birth and neurological disabilities in survivors. This paper adds several reports of results regarding the TRAP sequence using fetal treatment. I would like to provide some comments that I feel could improve the quality of the paper.
Methods
1. The application criteria for IFL and CO should be mentioned, since you compare the results of both treatments. How IFL and CO are each selected would necessarily affect the outcomes of these treatments. This point is crucial for this study.
2. Regarding the neurological assessment, the definition of developmental impairment should be addressed. Is it based on the DQ score or major findings at checkups? This definition is important for assessing the long-term outcomes.
3. You appear to have treated only MCDA twins, not monoamniotic (MCMA) twins with TRAP. Did you exclude cases of MCMA?
Results
I recommend that the authors show more results in the tables to improve the quality of the paper.
1. Table 1 shows a few basic characteristics and outcomes.
In addition to the GA at intervention and cardiac failure status, a number of characteristics are recommended to be shown in Table 1, including the GA at first presentation, MVP or polyhydramnios, acardiac mass size (e.g. abdominal circumference ratio of the acardiac/pump twin), operation time, total laser energy, procedure failure, and repeated procedures.
2. Outcomes: In addition to IUFD, the GA at birth, and rate of preterm delivery <34 weeks, a number of representative outcomes should be shown in Table 2, including the rate of infant death, survival rates, pPROM, birth weight, and neurological deficits.
Discussion
1. 2nd paragraph, line 157-165: The major point of this paragraph is unclear.
2. Strengths, line 182-184: Neurological data without a clear definition cannot be considered a strength. The paragraph on strengths and limitations usually appears in the last section of the Discussion.
3. RFA: IFL and RFA are the most frequently used interventions. Most centers in Europe and Latin America prefer IFL, whereas RFA is preferred in North America (Fetal Diagn Ther 2023 doi: 10.1159/000531791). It is recommended that RFA be discussed appropriately.
Line 214-215: “Several reports proved RFA in TRAP as a safe and effective method of treatment with no neurological impairment in the survivors [19,20].” However, references 19 and 20 are not papers on RFA but rather papers on IFL. Proper references, including the following, should be cited: The overall survival rate of pump twins treated with RFA was 81%-85% (34/40) (Fetal Diagn Ther 2014: 267-279, Prenat Diag 2016: 437-443). The incidence of neurodevelopmental delay after RFA was 0% (0/27) (Prenat Diag 2021: 1-7). Text in the Discussion comparing these RFA data should clarify the characteristics of each type of treatment.
Author Response
Outcome of monochorionic diamniotic twins with twin reversed arterial perfusion sequence: interstitial laser versus endoscopic cord occlusion.
We would like to thank the reviewer for the insightful comments that improved the presentation and quality of our paper. We have performed additional analyses and rewritten some parts of the manuscript to address issues raised by the reviewer. In the revised manuscript, all changes in the text and all new text fragments are marked in yellow. Please find below our point-by-point response to the reviewers’ comments.
- The application criteria for IFL and CO should be mentioned, since you compare the results of both treatments. How IFL and CO are each selected would necessarily affect the outcomes of these treatments. This point is crucial for this study.
Response: Thank you. In our study we offered intrauterine intervention in all cases diagnosed with TRAP sequence. The timing of the procedure was mainly dependent on the timing of diagnosis and referral. Intrafetal laser was a preferred technique in cases diagnosed and referred to the centers before 22 weeks gestation. Laser coagulation of the umbilical cord was performed in cases referred to the centers after 22 weeks gestation. The choice of the technique was also dependant on the the technical issues which included: position of the placenta, position of the TRAP fetus and patient’s body mass index (BMI).
The comment is now added in the lines 94-99.
- Regarding the neurological assessment, the definition of developmental impairment should be addressed. Is it based on the DQ score or major findings at checkups? This definition is important for assessing the long-term outcomes.
Response: Thank you. In our study developmental impairment was reported by the parents via telephone contact. The disabilities were established by the local pediatrician during regular national visit. The diagnosis of certain disabilities was also confirmed by neurologist. Unfortunately, we couldn’t perform developmental assessment of children in our center because of demographic varieties, different age of children and logistic disabilities.
The comment in now added in the lines 107-109.
- You appear to have treated only MCDA twins, not monoamniotic (MCMA) twins with TRAP. Did you exclude cases of MCMA?
Response: Thank you. Yes, we excluded MCMA twins.
- Table 1 shows a few basic characteristics and outcomes.
In addition to the GA at intervention and cardiac failure status, a number of characteristics are recommended to be shown in Table 1, including the GA at first presentation, MVP or polyhydramnios, acardiac mass size (e.g. abdominal circumference ratio of the acardiac/pump twin), operation time, total laser energy, procedure failure, and repeated procedures.
Response: Thank you. The table has been completed accordingly.
- Outcomes: In addition to IUFD, the GA at birth, and rate of preterm delivery <34 weeks, a number of representative outcomes should be shown in Table 2, including the rate of infant death, survival rates, pPROM, birth weight, and neurological deficits.
Response: Thank you. The table has been completed accordingly.
- 2ndparagraph, line 157-165: The major point of this paragraph is unclear.
Response: We thank the reviewer for comments. Our goal was to emphasize that all cases of TRAP should be offered an early intervention. The cited literature proves an early treatment as safe and effective. The above cited research explained a hidden mortality of the pump twin between diagnosis and treatment decision. There is no benefit in delaying therapeutic intervention. The paragraph is especially important in understanding benefits of early intervention. By using IFL before 16 weeks of gestation, we could decrease the pump twin’s mortality and improve development of surviving fetus.
- Strengths, line 182-184: Neurological data without a clear definition cannot be considered a strength. The paragraph on strengths and limitations usually appears in the last section of the Discussion.
Response. We thank the reviewer for comments. The definition of developmental impairment was added to material and methods section. Information about neurological status of children born after TRAP management is limited due to rarity of this condition. However, describing neurological status of the children born after intrauterine interventions rarely occur in literature, our data could be beneficial in the treatment optimization. We carefully reviewed the literature on fetal surgery in TRAP sequence and there was only one paper that mentioned long term outcomes of the survivors. Our paper adds new data on this important issue in fetal surgery therefore we consider it as a strength of the study.
- RFA: IFL and RFA are the most frequently used interventions. Most centers in Europe and Latin America prefer IFL, whereas RFA is preferred in North America (Fetal Diagn Ther 2023 doi: 10.1159/000531791). It is recommended that RFA be discussed appropriately. Line 214-215: “Several reports proved RFA in TRAP as a safe and effective method of treatment with no neurological impairment in the survivors [19,20].” However, references 19 and 20 are not papers on RFAbut rather papers on IFL. Proper references, including the following, should be cited: The overall survival rate of pump twins treated with RFA was 81%-85% (34/40) (Fetal Diagn Ther 2014: 267-279, Prenat Diag 2016: 437-443). The incidence of neurodevelopmental delay after RFA was 0% (0/27) (Prenat Diag 2021: 1-7). Text in the Discussion comparing these RFA data should clarify the characteristics of each type of treatment.
We thank the reviewer for comments. RFA is commonly use in North America, however in our record there were no cases treated with RFA. In the present study we wanted to compare long-term outcomes of treatment and highlighted that there is not much data about neurological development after IFL even if this method was commonly used. RFA could be proper intervention options, but the study by Pagani et al. which compares RFA and intrafetal laser highlights higher incidents of pPROM before 32 weeks of gestation when using RFA (22 vs 7% p=0.045). Future large data researches are needed to evaluate the therapy.
The study was revised as indicated (lines 229-232) with addition the RFA treatment data, however in the discussion we wanted to focus more on comparing IFL and CO methods. References have been corrected accordingly.
Reviewer 2 Report
The study aimed to describe and compare short and long-term outcomes of MC pregnancies with TRAP sequence treated with two different techniques: interstitial fetal laser versus endoscopic cord occlusion. The point of the discussion is very crucial for thinking about the intervention for TRAP sequence. It is very valuable and important data. In particular, the outcome of long prognosis should be reported. But further information and discussion are needed. Just the information they wrote could not say this conclusion.
-Preterm delivery is associated with the gestational week of the intervention and the maternal characteristics. And how the intervention was done (operation time, difficulty,,,) could also affect it. Are there any significant differences about them between IFL and CO? It should be indicated clearly.
-First of all, what was the indication of the fetal surgery for these cases? How was the timing of intervention decided on each case? The prognosis of pump twin is not only associated with the gestational week of delivery, but also if the acardiac twin get greater, if the doppler waveform of the pump twin is normal before and after intervention, if the acardiac twin have rudimentary heart,,, and so on. Further analysis is needed.
-Not only the cases with neurological disabilities are important. The cases without them are also very important. The authors need to provide further information about these cases. As the authors write, the number of cases is small. The important thing is not statistical analysis, but reviewing each case.
Author Response
interstitial laser versus endoscopic cord occlusion.
We would like to thank the reviewer for the insightful comments that improved the presentation and quality of our paper. We have performed additional analyses and rewritten some parts of the manuscript to address issues raised by the reviewer. In the revised manuscript, all changes in the text and all new text fragments are marked in yellow. Please find below our point-by-point response to the reviewers’ comments.
Reviewer 2
- Preterm delivery is associated with the gestational week of the intervention and the maternal characteristics. And how the intervention was done (operation time, difficulty,,,) could also affect it. Are there any significant differences about them between IFL and CO? It should be indicated clearly.
Response: We thank the reviewer for comments. Detailed information about type of intervention was added to the section “Material and Methods”. We have analyzed operation data and outcomes of pregnancies after TRAP management and we did not find any correlations between factors that can influence the course of pregnancy. We carefully analyzed early IUFD cases but none of factors was related with poor pregnancy outcomes. Additional data on the study group characteristics and pregnancy outcomes have been added in Tables 1 and 2.
- First of all, what was the indication of the fetal surgery for these cases? How was the timing of intervention decided on each case? The prognosis of pump twin is not only associated with the gestational week of delivery, but also if the acardiac twin get greater, if the doppler waveform of the pump twin is normal before and after intervention, if the acardiac twin have rudimentary heart,,, and so on. Further analysis is needed.
Response: We thank the reviewer for comments. We offer intrauterine procedure in all cases diagnosed with TRAP irrespective of the presence or absence of the cardiac overload. The timing of the procedure is therefore mainly dependent on the timing of the diagnosis and referral. There was a difference in the rate of cardiac overload and pump twin mass at the time of the procedure between the two groups however it hasn’t reached statistical difference. Additional data on this issue has been added in Tables 1 and 2 as well as in the lines 94-99.
- Not only the cases with neurological disabilities are important. The cases without them are also very important. The authors need to provide further information about these cases. As the authors write, the number of cases is small. The important thing is not statistical analysis, but reviewing each case.
Response: We thank the reviewer for comments, we agree with your opinion. The long-term follow-up is crucial in the assessment of treatment effectiveness and safety. In the presented study we contacted parents and/or pediatricians of each patient in order to gain information about children’s development. We focused on reimbursed newborn screening data including hearing examination, testing for congenital hypothyroidism, nervous system control and eye examination, phenylketonuria test, testing for cystic fibrosis. All children from IFL and most from CO group had proper neurological and hearing development. Three children in the group treated with CO had some disabilities as indicated in the discussion section. In the presented study we focused on describing cases with impairments more than comparing with each other. Additional information on neurological data analyzed in the study in now added in the lines 107-110.
Round 2
Reviewer 1 Report
Although I can clearly see that the authors have put a lot of effort into responding to my comments, especially with the tables, there are still several points that need to be resolved, as noted below:
Line 94-99: “The choice of the technique was depandant on multiple factors.” (line 94) This sounds strange. You might mean “The technique of intervention was chosen for several reasons” “The choice of the technique was also dependent on the technical issues, which included: the position of the placenta, position of the TRP fetus, and the patient’s body mass index (BMI).” (line 97/99) This is unclear. A clear description is required. For what position of the placenta or TRAP fetus and what BMI was IFL or CO preferred?
Line 229-330: “The overall survival rate of the pump treated with RFA was 81-85% (34/40)” These data are derived from papers on Fetal Diagn Ther 2014: 267-279 and Prenat Diag 2016: 437-443, which are not cited in this paper. According to reference 14, the sentence could be “The pooled estimated survival rate of the pump treated with RFA was 80.8% (72.3-87.1).” Again, proper references should be cited.
“The choice of the technique was depandant on multiple factors.” (line 94) This sounds strange.
Author Response
versus endoscopic cord occlusion.
We would like to thank the reviewers for their insightful comments that improved the presentation and quality of our paper.
Reviewer 1
Thank you. The manuscript was revised as indicated.
Reviewer 2 Report
I think that it gets better now by adding more data. If possible, it would be better to provide more data on cases without neurological disabilities or about Doppler waveform on each cases. However, even if authors don't have these data, this manuscript could be reported as one of important results.
I would like to suggest to make some corrections about new data.
- In Tables, standard deviation should be listed in data of mean.
-Statistical analysis should be conducted on data of mean or median as well.
Author Response
Outcome of monochorionic diamniotic twins with twin reversed arterial perfusion sequence: interstitial laser versus endoscopic cord occlusion.
We would like to thank the reviewers for their insightful comments that improved the presentation and quality of our paper.
Reviewer 2
We did an analysis of cases without neurological disabilities and found no factors related with neurological status. We did detailed description of children with neurological problems to emphasize factors which could influence long-term outcomes.
We added standard deviation and p-value as indicated.